# Smarter Sampling for LLM Judges: Reliable Evaluation on a Budget

## Abstract

Large language models (LLMs) are increasingly employed as judges for scalable evaluation of AI systems, where an LLM is prompted to assess the outputs of another model. This approach is particularly valuable for tasks with non-verifiable answers, but its reliability ultimately depends on alignment with human judgments. Because human annotations are expensive and time-consuming, especially in domains that demand expert knowledge such as clinical text generation, it is essential to reduce annotation effort while maintaining accurate estimates of judge reliability. In this work, we study the problem of estimating the intraclass correlation coefficient (ICC) between LLM judges and humans under limited annotation budgets. We derive Chernoff bounds on the estimation error, providing theoretical guarantees on sample requirements and reducing sample size requirements by an average of 18% compared to the baseline. Building on this, we propose and evaluate 6 sampling strategies designed to identify the most informative examples for annotation. Experiments on 4 diverse real-world datasets demonstrate that our methods yield narrower confidence intervals and achieve relative improvements of 5.5%–31% in ICC precision over random sampling baselines.

## 1 Introduction

Large language models (LLMs) are increasingly used for text generation tasks, but their rapid adoption has outpaced our ability to evaluate them at scale (Thirunavukarasu et al., 2023; Meyer et al., 2023; Yuan et al., 2021; Celikyilmaz et al., 2021). Human evaluation remains the gold standard but is slow and expensive, particularly in domains like healthcare. To address these limitations, the *LLM-as-a-judge* framework, in which one LLM evaluates the outputs of another model, has emerged as a promising alternative (Gu et al., 2025). Recent work has explored this direction through human-labeled benchmarks (Dubois et al., 2024) and reference-free evaluation methods (Tan et al., 2025). However, the effectiveness of an LLM judge ultimately depends on alignment with human judgments.

Human-annotated scores are typically treated as the gold standard when evaluating a target LLM. However, human evaluation pipelines are slow and expensive, particularly in domains requiring expert annotations such as healthcare (Liang et al., 2022; Kiela et al., 2021; Arndt et al., 2017). This creates a bottleneck for scalable evaluation: the cost of human labels directly limits the feasibility of benchmarking and alignment studies. To quantify judge reliability, we follow prior work (Bedi et al., 2025; Li et al., 2024a; Croxford et al., 2025) and adopt the intra-class correlation coefficient (ICC) as a measure of agreement between LLM judges and humans. This leads to two central questions: *(i) How many human annotations are needed to accurately estimate ICC? (ii) How can we choose the most informative samples to minimize annotation cost?*

To address the first question, we derive a concentration inequality for ICC estimation using the classical Chernoff bound (Chernoff, 1952) on an asymptotic normal approximation of the ICC distribution. This yields a lower bound on the number of annotations required to guarantee with high probability that the empirical ICC is within $\varepsilon$ of the population ICC.

For the second question, we empirically study how annotation efficiency can be improved through principled sample selection. We frame subset selection for human annotation as a core-set selection problem, assuming access to a fully LLM-annotated dataset. Building on ideas from statistical sampling (Cochran, 1977), clustering (Lloyd, 1982), and active learning (Settles & Craven, 2008),

Figure 1: **Overview of the LLM judge evaluation framework and our approach.** Given text samples T to evaluate, we obtain inexpensive scores G from an LLM judge and expensive scores H from human annotators. Our work addresses two key questions: (i) *How many human annotations are needed to reliably estimate agreement between G and H?* We derive a Chernoff bound-based sample size requirement (3.2). (ii) *Which samples should be selected for human annotation?* We propose and evaluate 6 sampling strategies, with cluster-based selection achieving up to 31% relative improvement in ICC estimation precision compared to random sampling under limited annotation budgets (5).

we compare several sampling strategies against random selection and evaluate their impact on ICC estimation under limited annotation budgets.

Given the pressing need to evaluate LLM judges under limited annotation budgets, we propose a theoretical framework for ICC estimation and principled sampling strategies. **Our contributions:**

1. We derive a Chernoff bound–based concentration inequality for ICC estimation, providing theoretical guarantees on the number of annotations required for reliable LLM judge evaluation.

2. We conduct a systematic empirical study of sampling strategies for annotation efficiency across 4 diverse real-world datasets spanning 15 axes of assessment.

3. We show that principled selection consistently outperforms random sampling under tight budgets ($\leq 5\%$ of data), with relative gains ranging from 5.5% to 31% in ICC estimation precision.

## 2 RELATED WORK

Prior work can be grouped into three areas: (i) the use of LLMs-as-judges for scalable evaluation, (ii) statistical and sampling methods for reliability and label efficiency, and (iii) evaluation metrics for measuring agreement between human and model judgments.

**LLM Evaluation Methods.** LLMs are increasingly used as automatic evaluators of other AI systems, offering a scalable alternative to costly human assessment. Early studies compared LLM judgments with human annotations on tasks such as summarization, dialogue, and reasoning (Gilardi et al., 2023; Zheng et al., 2023). More recent frameworks, such as AlpacaEval 2.0 and Arena-Hard, extend this line of work by integrating human and LLM judgments or introducing more challenging comparative tasks (Li et al., 2023; 2024b). Other recent frameworks, like the G-Eval framework (Liu et al., 2023) or the DHP benchmark (Wang et al., 2024), evaluate the quality of generated texts with a form-filling paradigm and generate evaluation scores on a Likert scale Joshi et al. (2015). Additional research has been conducted on using active learning techniques for evaluation, where informative and diverse examples are selectively labeled to guide prompt optimization in LLM-as-Judge systems (Zhen et al., 2025). Benchmarks such as JUDGE-BENCH (Bavaresco et al., 2024), provide an overview of LLM-judge reliability and consistency with human judgments. Different metrics have been proposed to assess the reliability of judge LLMs against human labels, drawing on classic statistics literature, including the intraclass correlation coefficient (ICC), Cohen's kappa, and Cronbach's alpha (Shrout & Fleiss, 1979; Cohen, 1960; Cronbach, 1951). These approaches highlight the importance of quantifying judge reliability through measures of accuracy, agreement, or correlation

with human preferences. Among them, the ICC has become a standard for evaluating continuous or ordinal judgments (Bedi et al., 2025; Croxford et al., 2025; Li et al., 2024a). We adopt ICC as the primary metric for both our theoretical and experimental analyses.

**Statistical foundations.** Building on this need for reliability, the statistical foundation for ICC estimation and sample size planning is well-grounded (Fisher, 1925; Shrout & Fleiss, 1979; Bonett, 2002; Zou, 2012; Giraudeau & Mary, 2001). The bulk of previous work has focused on statistical error in estimation of sample ICC, providing bound on confidence interval half-width. Common practice in a variety of statistical inference and estimation applications leverages concentration inequality frameworks for theoretical guarantees (Zhang & Chen, 2020; Wainwright, 2019; Koltchinskii & Lounici, 2017). More recent work has utilized concentration inequality frameworks to provide estimation bounds for Pearson's correlation coefficient (also termed sample or *inter*-class correlation coefficient) (Salnikov, 2024). Similarly, we port concentration inequality tools for sample complexity bounds in ICC. As in literature in confidence interval bounding, we use Fisher's asymptotic normal approximation of the density for ICC.

**Sample Selection Strategies.** Reducing reliance on human labels has long been studied in active learning and sample-efficient evaluation, where the challenge is to identify the most informative examples (Settles, 2009; Wei et al., 2022). One common approach is *Query-By-Committee* (QBC), which selects samples that maximize disagreement, thereby prioritizing uncertain examples (Seung et al., 1992). Extensions such as *stratified QBC* combine disagreement-based selection with stratified sampling principles from classical statistics (Cochran, 1977), ensuring both informativeness and balanced subgroup coverage. Representativeness is often pursued through clustering-based approaches such as $k$-means (Lloyd, 1982; Dong et al., 2025), which partition the data before sampling. Similarly, *stratified sampling* ensures proportional representation across predefined subgroups, reducing variance and improving estimate reliability. *Variance-maximization* based sampling approaches are based in experimental design theory (Fedorov, 1972), which motivates selection strategies that maximize the variability of the chosen item set, improving the informativeness of smaller subsets. *Density-based approaches* from statistics and active learning (Silverman, 1986; Nguyen & Smeulders, 2004) balance typical high-density examples with low-density outliers, promoting subsets that capture both central trends and rare but important cases.

Taken together, our work connects these threads by (i) providing theoretical guarantees for ICC estimation under limited annotation budgets and (ii) by designing principled sampling strategies tailored to LLM-judge evaluation.

## 3 THEORETICAL FOUNDATIONS

### 3.1 BACKGROUND ON INTRA-CLASS CORRELATION COEFFICIENT

Intra-class Correlation Coefficient (ICC) was originally proposed by Fisher (1925) as an extension to *interclass* correlation coefficient (Pearson's correlation coefficient (PCC)), and measures the extent to which the total variance in observed data is due to differences between groups, rather than within groups. In this perspective, the ICC is understood within the analysis of variance (ANOVA) framework. As opposed to PCC, the data are pooled in the mean calculation.

In the generic version of our use case, ICC is considered a measure that quantifies inter-rater reliability between $k$ raters on $n$ subjects, first introduced as an application of the metric in Shrout & Fleiss (1979). ICC measures reliability by decomposing the total variance in human evaluations into between-subjects variance and within-subjects error variance. The ICC determines the reliability of ratings by comparing the variability of different ratings of the same individuals to the total variation across all ratings and all individuals. As we only consider two raters, the human and the LLM, we consider the case $k = 2$.

Analogously, modern ICC estimators derive ICC through the random effects model framework. In the random effects model, $X_{ij}$, rating $j$ on subject $i$, $i \in [n], j \in [k]$, is modeled as

$$X_{ij} = \mu + \alpha_i + c_j + \varepsilon_{ij}$$

such that $\mu$ is an unobserved overall mean, $\alpha_i$ is an unobserved random effect shared by all ratings on subject $i$, $c_j$ is an unobserved random effect shared by all subject ratings by rater $j$, and $\varepsilon_{ij}$ is an unobserved noise term. Each class of terms is assumed to be respectively identically distributed

with expected value $0$, and the terms are assumed to be uncorrelated. For certain random effects models, either $\alpha_i$ or $c_j$ is neglected or considered fixed. We refer to Liljequist et al. (2019) for a comprehensive overview of ICC definitions and derivations relating classical estimators to the random effects model. See table in appendix A.1 for reproduced formulas.

In our specific use case, we use a two-way consistency average, i.e. $\mathrm{ICC}(3, k)$ as this formulation treats *raters* as fixed effects, (i.e. $c_j$ is fixed), meaning the same evaluation panel assesses all LLM outputs, and estimates reliability for the average rating across $k$ evaluators rather than individual rater consistency. The numerator $(MS_R - MS_E)$ captures the true variance between different LLM responses after removing measurement error, while the denominator represents the total variance in averaged ratings, making $\mathrm{ICC}(3, k)$ particularly sensitive to systematic differences in how evaluators rate different model outputs while accounting for random measurement error within the evaluation process. With random effects model for $\mathrm{ICC}(3, k)$, the population ICC

$$\rho = \frac{\sigma_\alpha^2}{\sigma_\alpha^2 + \sigma_\varepsilon^2/k}$$

We utilize the associated formula as the ICC metric for our experiments due to the appropriateness of the setting, random effects model, and use in previous empirical work (Bedi et al., 2025; Croxford et al., 2025; Li et al., 2024a). In our theoretical analysis, we provide bounds with $\mathrm{ICC}(3, 1)$, as the expression resembles Fisher's original proposal for ICC and follows previous theoretical work (Zou, 2012; Bonett, 2002; Giraudeau & Mary, 2001). Under $\mathrm{ICC}(3, 1)$, the associated random effects model dictates that the population ICC

$$\rho = \frac{\sigma_\alpha^2}{\sigma_\alpha^2 + \sigma_\varepsilon^2}$$

This is additionally the more commonly stated population ICC. Note that $\mathrm{ICC}(3, k)$ measures the reliability of the measurement as the average of $k$ raters, whereas $\mathrm{ICC}(3, 1)$ measures the reliability of each single measurement. In the case where $k = 2$, these do not differ greatly. As previously stated, we consider $k = 2$ only in both our empirical and theoretical results.

### 3.1.1 PREVIOUS WORK ON BOUNDS FOR ICC

Previous work in bounding error in ICC estimation focuses on bounding the half-width of a confidence interval (Zou, 2012; Bonett, 2002; Giraudeau & Mary, 2001). In the more recent of these works, used empirically (Bedi et al., 2025), Zou (2012) aims to determine the required sample size for estimating the intraclass correlation coefficient (ICC) with a desired $(1 - \alpha)100\%$ two-sided confidence interval half-width, $\omega$ and pre-specified assurance probability, $1 - \beta$. Thus, Zou (2012) sets

$$1 - \beta = \Pr\left[z_{\alpha/2}\sqrt{\mathrm{Var}(\hat{\rho})} \leq \omega\right]$$

to obtain a minimum bound on the required sample size such that the half-width of the confidence interval remains within the desired width with probability $1 - \beta$, where $\hat{\rho}_n$ denotes the sample $\mathrm{ICC}(3, 1)$ on $n$ samples. Similarly, the variance $\mathrm{Var}$ here is assumed to be the sample variance as opposed to population variance, so $\sqrt{\mathrm{Var}(\hat{\rho})}$ is simply the sample standard deviation $\hat{\sigma}_{\hat{\rho}}$, where the $(1-\alpha)100\%$ two-sided confidence (Wald) interval is given by $\hat{\rho} \pm z_{\alpha/2}\hat{\sigma}_{\hat{\rho}}$, thus clearly the half-width of the confidence interval is $z_{\alpha/2}\hat{\sigma}_{\hat{\rho}}$.

Following this, Zou (2012) shows that for $\mathrm{ICC}(3, 1)$ with two raters (human and LLM), the required sample size is:

$$n = 1 + \left[\frac{Az_{\alpha/2} + \sqrt{A^2z_{\alpha/2}^2 + 4\omega z_{\alpha/2}z_\beta A|B|}}{\omega\sqrt{2k(k-1)}}\right]^2$$

$$= 1 + \left[\frac{(1-\rho^2)z_{\alpha/2} + \sqrt{(1-\rho^2)^2z_{\alpha/2}^2 + 8\omega z_{\alpha/2}z_\beta(1-\rho^2)|\rho|}}{2\omega}\right]^2$$

where the latter line comes from plugging in $k = 2$, $A = (1 - \rho)[1 + (k - 1)\rho] = 1 - \rho^2$, $B = k - 2 + 2\rho - 2k\rho = -2\rho$ (thus $|B| = 2|\rho|$), $\omega$ is the desired half-width, and $z_\gamma$ denotes the upper $\gamma$ quantile of the standard normal distribution.

In the next section, we translate the question of bounding the half-width of the confidence interval with high probability to that of bounding the difference between sample and population ICC with high probability – a concentration inequality framework. The two frameworks are translatable, but our concentration inequality framework is ideal for *directly* bounding the error with high probability, and requires fewer assumptions than those present in the calculation of, for instance, a Wald interval, as used in Zou (2012).

### 3.2 An Approximate Chernoff bound for Intra-class Correlation Coefficient

Here, we derive a straightforward concentration inequality for the intra-class correlation coefficient (ICC). Given annotation distributions, $H$ and $G$, for human and LLM-generated, respectively, we assume a bivariate normal joint distribution, such that $H_i, G_i \sim \mathcal{N}(\mu, \Sigma)$ independently and identically distributed (i.i.d.), and $i \in \{1, \ldots, n\}$, where $n$ number of samples. Denote the population ICC between $H$ and $G$ as $\rho$ and the observed (sample) ICC at $n$ samples, $\hat{\rho}_n$.

Fisher (1925) showed that given sufficient sample size $n$, and $\rho$ not too close to its boundary $[-1, 1]$, the distribution of $\hat{\rho}_n$ approaches normal asymptotically. In particular, the distribution of $\hat{\rho}_n$ approaches a Gaussian distribution with variance $\frac{(1-\rho^2)^2}{n-1}$. Therefore, we can use the Chernoff bound technique to derive a simple concentration inequality for the intra-class correlation coefficient.

We therefore obtain the following lemma.

**Lemma 1** (Chernoff bound for approximate ICC). *Let $H$, $G$ be two random variables of interest, and assume independently and identically distributed samples $H_i, G_i \sim \mathcal{N}(\mu, \Sigma)$ sampled from a bivariate normal distribution. Let $\rho$ denote the population ICC, and $\hat{\rho}_n$ denote the sample ICC, as defined in Section 3.1. Given desired bound parameter $\varepsilon > 0$, $n$ sufficiently large such that CLT holds, and $|\rho|$ not close to 1,*

$$\Pr[|\hat{\rho}_n - \rho| \geq \varepsilon] \lesssim 2 \exp\left(-\frac{(n-1)\varepsilon^2}{2(1-\rho^2)^2}\right)$$

*Therefore, given $\delta > 0$, with probability $1 - \delta$, the sample and population ICC are guaranteed to be $\varepsilon$-close if*

$$n \gtrsim 1 + \frac{2(1-\rho^2)^2}{\varepsilon^2} \log\left(\frac{2}{\delta}\right)$$

*See proof in Appendix A.2*

We provide a comparison between our Chernoff bound and the bound from Zou (2012) in Table 1. As the bound frameworks technically bound two different events, we must translate between the relevant parameters. See Appendix A.3 for more details. Effectively, $\varepsilon = \omega$ and $\delta = 1 - (1 - \alpha) \cdot (1 - \beta)$. As shown in Table 1, the proposed Chernoff bound is more sample efficient, providing relative gains over the current baseline of up to 25.0% for $\rho = 0.8$ and an average of 18.3% over all $\rho$ values[1].

## 4 Methods

### 4.1 Problem Formulation

We study the problem of evaluating LLM judges under limited annotation budgets. Assume we have a set of items $\mathcal{X} = \{x_1, \ldots, x_n\}$ for evaluation, with associated inexpensive labels $G = \{g_i\}$ from an LLM judge. Similarly, there exists a set of gold labels $H = \{h_i\}$ from humans of which we are only able to collect some subset of size $b$. Reliability is measured using the Intraclass Correlation Coefficient (ICC$(3, k)$), which captures absolute agreement. Further information regarding the calculation of ICC scores can be found in Appendix A.1.

---

[1]Note that cases where the derived sample size is below 30, this violates the assumptions required for both bounds (that n is sufficiently large according to CLT).

| | | Confidence Interval Parameters | | | Chernoff Bound Parameters | | |
|---|---|---|---|---|---|---|---|
| $\rho$ | $\alpha$ | $\omega$, half width | $\beta$ | Zou (2012) N | $\varepsilon$ | $\delta$ | (Ours) N |
| 0.6 | 0.05 | 0.1 | 0.5 | 158 | 0.1 | 0.525 | **111** |
| 0.6 | 0.05 | 0.15 | 0.5 | 71 | 0.15 | 0.525 | **50** |
| 0.6 | 0.05 | 0.2 | 0.5 | 40 | 0.2 | 0.525 | **28** |
| 0.6 | 0.05 | 0.1 | 0.2 | 183 | 0.1 | 0.240 | **175** |
| 0.6 | 0.05 | 0.15 | 0.2 | 87 | 0.15 | 0.240 | **78** |
| 0.6 | 0.05 | 0.2 | 0.2 | 52 | 0.2 | 0.240 | **44** |
| 0.6 | 0.05 | 0.1 | 0.1 | **195** | 0.1 | 0.145 | 216 |
| 0.6 | 0.05 | 0.15 | 0.1 | **95** | 0.15 | 0.145 | 97 |
| 0.6 | 0.05 | 0.2 | 0.1 | 58 | 0.2 | 0.145 | **55** |
| 0.7 | 0.05 | 0.1 | 0.5 | 101 | 0.1 | 0.525 | **71** |
| 0.7 | 0.05 | 0.15 | 0.5 | 45 | 0.15 | 0.525 | **32** |
| 0.7 | 0.05 | 0.2 | 0.5 | 26 | 0.2 | 0.525 | **18** |
| 0.7 | 0.05 | 0.1 | 0.2 | 123 | 0.1 | 0.240 | **111** |
| 0.7 | 0.05 | 0.15 | 0.2 | 60 | 0.15 | 0.240 | **50** |
| 0.7 | 0.05 | 0.2 | 0.2 | 37 | 0.2 | 0.240 | **29** |
| 0.7 | 0.05 | 0.1 | 0.1 | **134** | 0.1 | 0.145 | 138 |
| 0.7 | 0.05 | 0.15 | 0.1 | 67 | 0.15 | 0.145 | **62** |
| 0.7 | 0.05 | 0.2 | 0.1 | 42 | 0.2 | 0.145 | **35** |
| 0.8 | 0.05 | 0.1 | 0.5 | 51 | 0.1 | 0.525 | **36** |
| 0.8 | 0.05 | 0.15 | 0.5 | 23 | 0.15 | 0.525 | **16** |
| 0.8 | 0.05 | 0.2 | 0.5 | 13 | 0.2 | 0.525 | **10** |
| 0.8 | 0.05 | 0.1 | 0.2 | 68 | 0.1 | 0.240 | **56** |
| 0.8 | 0.05 | 0.15 | 0.2 | 35 | 0.15 | 0.240 | **25** |
| 0.8 | 0.05 | 0.2 | 0.2 | 22 | 0.2 | 0.240 | **15** |
| 0.8 | 0.05 | 0.1 | 0.1 | 77 | 0.1 | 0.145 | **69** |
| 0.8 | 0.05 | 0.15 | 0.1 | 40 | 0.15 | 0.145 | **31** |
| 0.8 | 0.05 | 0.2 | 0.1 | 26 | 0.2 | 0.145 | **18** |

Table 1: **The proposed Chernoff bound tighter in the required number of human annotations.**
Relative to the interval-based bound of Zou (2012), our Chernoff bound achieves tighter bound by
18.3% in required sample size, with a median reduction of 21.6%. The gains are more pronounced
at higher correlations: average improvements are 12.4% for $\rho = 0.6$, 17.6% for $\rho = 0.7$, and 25.0%
for $\rho = 0.8$. The bold entries in the table indicate the lower sample complexity between the two
bounds.

Given a budget $b < n$, we seek a subset $S^* \subseteq \mathcal{X}$ of size $b$ such that the ICC $\hat{\rho}_b$ computed on
$(H_{S^*}, G_{S^*})$ closely approximates the ICC $\hat{\rho}_n$ on the full dataset $(H, G)$:

$$S^* = \arg \min_{S \subseteq \mathcal{X}, |S| = b} |\hat{\rho}_b(H_S, G_S) - \hat{\rho}_n(H, G)|.$$

We assume access to $\mathcal{X}$ and $G$, and a single-batch labeling regime (rather than an *active* iterative
setting) such that labels in $H_S$ are obtained all at once for a chosen $S$. Thus, our subset selection
must rely solely on the items $\mathcal{X}$ and the inexpensive labels $G$.[2]

### 4.2 DATA PREPARATION

We provide empirical evaluations on 4 diverse real-world datasets: MSLR (Wang et al., 2023),
HANNA (Chhun et al., 2024), MedVAL (Aali et al., 2025), and SummEval (Fabbri et al., 2021).
Each dataset is annotated along multiple axes (e.g., faithfulness, accuracy, creativity) with a total
of 15 sets of human annotations by which to evaluate sampling performance, as shown in Table 2.
All datasets contain samples that are annotated by more than one human, creating a distribution
of continuous scores that requires the use of ICC over alternative ordinal or categorical agreement
metrics. We take the average score of a subject between total raters as the true score and compare
to the LLM judge score. All datasets use Likert scaling with a range of 1–5 (Joshi et al., 2015).
The evaluation axes span from more objective to more subjective scores, capturing a broad spectrum
of evaluation types. We truncate each dataset to contain 300 human-labeled samples and perform
evaluations on subsets of this annotation set. We perform the evaluations over 100 rollouts, each with
its own random seed to account for variance in the predicted ICC value from the random components
of the selection mechanisms (i.e., random selection, cluster initialization, etc.).

---

[2]As LLM-annotated data is inexpensive, we assume that $n$ is large, and $\hat{\rho}_n$ effectively acts as a stand-in
for the population ICC $\rho$ on the joint distribution $(H|\mathcal{X}, G|\mathcal{X})$, since we cannot access $\rho$ in our evaluation.
Similarly, we allow some abuse of notation such that $H = \{h_i\}_{i=1}^n$ and $G = \{g_i\}_{i=1}^n$ are assumed to capture
label distributions $H|\mathcal{X}$ and $G|\mathcal{X}$, respectively.

Table 2: Datasets, evaluation axes, and number of raters per datapoint.

| Dataset | Axes of Evaluation | Raters per datapoint |
|---------|--------------------|-----------------------|
| SummEval | Coherence, Consistency, Fluency, Relevance | 8 |
| HANNA | Relevance, Coherence, Empathy, Surprise, Engagement, Complexity | 3 |
| MedVAL | Safety | 1–3 |
| MSLR | Fluency, Population, Intervention, Outcome | 1–2 |

### 4.3 SAMPLING METHODS

We compare 6 strategies for selecting $S^*$ as shown in Table 3 against random sampling. Additional information regarding each algorithm is present in Appendix A.4

Table 3: Selection strategies for choosing $k$ items from cheap ratings $G$.

| Method | Description | Formula |
|--------|-------------|---------|
| Random | Uniformly sample $k$ items without replacement | $S_{\text{rand}} = \text{UniformSample}(\mathcal{N}, k)$ |
| Stratified | Partition ratings into $k$ quantile strata; sample one per stratum | $S_{\text{strat}} = \bigcup_{j=1}^{k} \text{UniformSample}(\text{Stratum}_j, 1)$ |
| QBC | Select items with largest inter-rater difference | $S_{\text{QBC}} = \arg\max_{\lvert S \rvert = k} \sum_{i \in S} \lvert g_i^{(1)} - g_i^{(2)} \rvert$ |
| Stratified QBC | Combine stratified $(k/2)$ and QBC $(k/2)$ | $S_{\text{sQBC}} = S_{\text{strat}}^{(k/2)} \cup S_{\text{QBC}}^{(k/2)}$ |
| Cluster | Choose item nearest to each K-means centroid | $S_{\text{clust}} = \{\arg\min_{i \in C_j} \lvert g_i - c_j \rvert\}_{j=1}^{k}$ |
| Maximum-Variation | Iteratively add items maximizing variance | $S_{t+1} = S_t \cup \{\arg\max_{i \notin S_t} \dots\}$ |
| Density-Based | Balance typical (high density) and outliers (low density) | $S_{\text{dens}} = S_{\text{high}} \cup S_{\text{low}}$ |

## 5 RESULTS

### 5.1 ESTIMATION ERROR BY SELECTION METHOD

As shown in Figure 2, the **cluster** method consistently achieves a lower estimation error than random selection. Across all datasets and tasks, cluster-based selection consistently provides more sample-efficient ICC estimation with $n_{expensive} \leq 30$, with up to 31% relative improvement over random sampling in settings with $\leq 5\%$ of annotation budget available as shown in Table 4. We see saturation of gains from cluster-based selection as the data budget ($n_{expensive}$) increases, as random sampling is better able to capture the true distribution of samples with increased data points. While some relative improvements are negative at larger $n_{expensive}$, the difference between estimation error from cluster-based selection and random selection decreases as $n_{expensive}$ increases, so these negative values represent small differences (absolute differences are present in Appendix Table 5). However, in data-scarce settings, cluster selection consistently allows for lower estimation error between predicted and true ICC values.

In addition to minimizing the estimation error between true and predicted ICC values, cluster-based selections are also more stable and have a demonstrably smaller confidence interval than equivalent random selection, as shown in Table 4. For datasets HANNA, MedVAL, and SummEval, the improvement in confidence interval width is larger than with the MSLR dataset. Similarly, the MSLR dataset also shows the smallest improvement against random in estimation error as well as confidence interval width. We attribute this to the true ICC for all samples in this dataset against the

judge model. The ICC of MSLR is $0.346$, while the ICC of HANNA, MedVAL, and SummEval, respectively, are $0.655, 0.716, 0.627$. This indicates that clustering selection provides the most benefit under the assumption of a minimum ICC and concordance between the judge model and human scoring. When the raters are dissimilar, the clustering method converges to perform similarly to random selection.

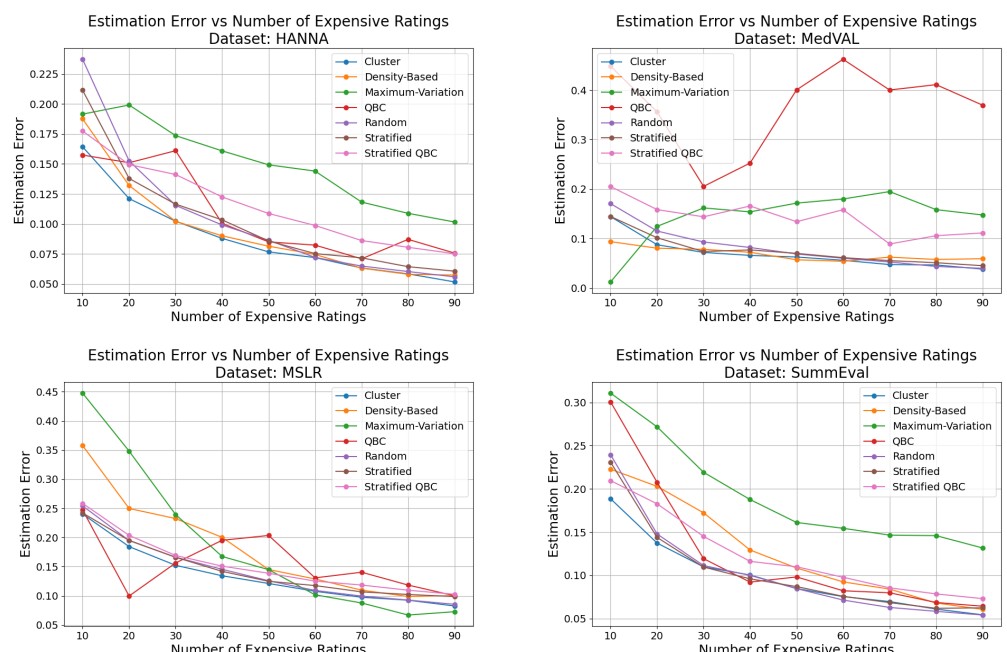

Figure 2: **Cluster-based sample selection for annotation leads to consistently lower estimation error.** We compare 6 methods of sample selection against a random baseline, showing that cluster-based approaches consistently perform at or above the precision level of random selection, providing improved estimations of true ICC values.

| $n_{\text{expensive}}$ | Mean ICC Improvement over Random (%) | | | | CI Width Improvement of Cluster over Random (%) | | | |
|---|---|---|---|---|---|---|---|---|
| | HANNA | MedVAL | MSLR | SummEval | HANNA | MedVAL | MSLR | SummEval |
| 10 | 31.0% | 15.0% | 5.5% | 21.0% | 6.4% | 32.4% | $-0.9\%$ | 24.1% |
| 20 | 21.0% | 24.0% | 5.4% | 7.1% | 7.8% | 18.3% | 0.1% | 21.9% |
| 30 | 11.0% | 23.0% | 8.4% | 1.4% | 7.1% | 13.9% | $-2.4\%$ | 18.4% |
| 40 | 11.0% | 20.0% | 7.6% | 0.0% | 2.6% | 9.7% | $-1.0\%$ | 14.6% |
| 50 | 11.0% | 9.0% | 3.7% | 0.0% | 4.9% | 9.4% | $-0.7\%$ | 14.0% |
| 60 | 0.0% | 6.8% | 1.3% | $-5.6\%$ | 2.9% | 2.1% | 0.0% | 11.3% |
| 70 | 2.5% | 10.0% | 2.0% | $-11.0\%$ | 3.5% | 8.0% | $-0.4\%$ | 9.1% |
| 80 | 3.2% | $-7.1\%$ | 0.0% | $-4.4\%$ | 1.5% | 5.9% | 0.0% | 7.8% |
| 90 | 7.5% | 4.2% | 3.4% | 0.0% | 1.8% | 4.5% | $-1.3\%$ | 6.2% |

Table 4: **Cluster-based sampling can decrease estimation error and improve confidence intervals in low data settings.** Relative improvement (%) of Cluster over Random for both average ICC (left block) and confidence interval width (right block). Negative values indicate cases where Cluster underperforms Random.

## 5.2 COVERAGE PERSPECTIVE ON CLUSTERING

As discussed in Section 4.1, we frame our task as selecting a subset of samples such that the empirical intra-class correlation coefficient (ICC) on $b$ annotated items, $\hat{\rho}_b$, closely approximates the ICC on the full dataset, $\hat{\rho}_n$. Since human labels are not available a priori, we cannot sample based on them directly. An optimal subset $(H_S, G_S)$ should preserve the distributional properties of the

full set $(H, G)$, specifically the variance structure that determines ICC. While the human-label variance components are inaccessible, the inexpensive labels provide a coarse but useful proxy. Thus, our goal is to select a subset $G_S$ whose distribution approximates that of $G$, effectively minimizing regret with respect to set coverage.

This motivates the use of clustering approaches to adequately capture set variance. A known problem in the context of set coverage is the k-Centers problem. Given a metric space $(\mathcal{X}, d)$, the objective of the k-Centers problem (Hakimi, 1964) is to find $k$ centers $\mathcal{C}^* = \{c_1, \ldots, c_k\} \subseteq \mathcal{X}$ such that

$$\mathcal{C}^* = \arg \min_{C \subseteq \mathcal{X}, |C|=k} \max_{x \in \mathcal{X}} \min_{c \in C} d(c, x)$$

Although this problem is NP-hard, approximate solutions yield strong coverage guarantees (Lim et al., 2005). While $k$-means clustering optimizes a different criterion (minimizing within-cluster variance rather than worst-case distance), it also provides effective coverage of the space (Wolf, 2011). By covering the inexpensive label space, $k$-means ensures that the selected subset spans the diversity of the dataset, which in turn helps recover the variance structure and ICC of the full population. Prior work has similarly leveraged clustering for diverse sampling in related contexts (Sener & Savarese, 2018).

## 6 DISCUSSION

As LLMs become central in various workflows, the ability to evaluate their outputs at scale becomes increasingly critical. LLM Judges emerge as the current state-of-the-art for scalable, non-verifiable evaluation, but these methods are only as good as they are aligned with human preferences. One way to quantify their correlation with human ratings is through the intra-class correlation coefficient (ICC). LLM judge outputs can be compared to gold-standard human outputs with ICC, but human outputs are expensive and time-intensive to obtain. We explore methods to reduce the human annotation burden.

We introduce a Chernoff bound that allows practitioners to derive an approximate bound on the minimum number of human annotations necessary to ascertain performance given a desired tolerance $\varepsilon$ and probabilistic guarantee $\delta$. This bound improves upon current baseline methods of sample size calculation by an average of 18.3% across parameter combinations. Additionally, we show that cluster-based selection consistently provides the most sample-efficient estimation of the true ICC value across a large range of potential "budget" ratios. The greatest improvements occur in low-budget settings, with relative improvement of 5.5% to 31% compared to random in settings where annotation budget is $\leq 5\%$ of total samples. This allows model practitioners to iterate on their LLM judge methodology with higher fidelity without wasting annotation budget. We compare the confidence interval size between randomly selected samples and samples selected using a cluster-based approach. We see that the cluster-based approach provides tighter confidence intervals in addition to the empirical results of more precise ICC estimates. This supports that in limited annotation settings, cluster-based selection of points to receive human annotation decreases ICC estimation error as well as produces narrower confidence intervals for the ICC estimate.

Future work can explore extensions beyond cluster-based methods to further reduce expensive annotation requirements. The utilization of the subject text as a part of the selection process could further act as a signal of diversity and coverage, thus allowing for improved sampling at lower budgets and exploring associated bounds with specific data selection mechanisms. Currently, the cluster method is most impactful under the key assumption that the LLM judge ratings are a reasonable proxy for human ratings. Further exploration should provide means to evaluate this assumption a priori, such that practitioners have a base understanding of what the alignment may be and whether these assumptions hold.

## ETHICS STATEMENT

This work develops methods for estimating the reliability of large language model (LLM) judges while reducing annotation costs. We only use publicly available datasets. We do not collect new human subject data, and our experiments do not involve personally identifiable or sensitive information. A potential risk of this research is that efficiency gains in estimating reliability could be

misapplied to justify reducing necessary human oversight, particularly in safety-critical domains such as healthcare, law, or education. We stress that our methods are intended to improve the rigor of evaluation and reduce redundant annotation effort, not to replace expert human judgment. Any deployment in high-stakes settings must retain appropriate levels of domain-specific human review.

## REPRODUCIBILITY STATEMENT

We have taken several steps to ensure the reproducibility of our results. All theoretical derivations are provided in full in the main text and appendix. For empirical evaluation, we specify all datasets, baselines, and evaluation metrics in detail. Sampling strategies and corresponding implementation details are described in the methods section. All datasets used are publicly available and require no additional pre-processing beyond what is documented. We also release all code required for reproducing our empirical results at the anonymized repository: Smarter_Sampling.

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

## A  APPENDIX

### A.1  INTRA-CLASS CORRELATION COEFFICIENT DEFINITIONS

See table below for reproduced formulas (Liljequist et al., 2019).

| Name | Notation | Rater Model | Use Case | Formula |
|------|----------|-------------|----------|---------|
| One-way single | ICC(1,1) | Random | Agreement of 1 random rater | $\dfrac{\text{MS}_\text{R} - \text{MS}_\text{E}}{\text{MS}_\text{R} + (k-1)\text{MS}_\text{E}}$ |
| One-way average | ICC(1,k) | Random | Agreement of average random raters | $\dfrac{\text{MS}_\text{R} - \text{MS}_\text{E}}{\text{MS}_\text{R}}$ |
| Two-way absolute single | ICC(2,1) | Random | Absolute agreement of 1 random rater | $\dfrac{\text{MS}_\text{R} - \text{MS}_\text{E}}{\text{MS}_\text{R} + (k-1)\text{MS}_\text{E} + \frac{k}{n}(\text{MS}_\text{C} - \text{MS}_\text{E})}$ |
| Two-way absolute average | ICC(2,k) | Random | Absolute agreement of average raters | $\dfrac{\text{MS}_\text{R} - \text{MS}_\text{E}}{\text{MS}_\text{R} + \frac{1}{n}(\text{MS}_\text{C} - \text{MS}_\text{E})}$ |
| Two-way consistency single | ICC(3,1) | Fixed | Consistency of 1 fixed rater | $\dfrac{\text{MS}_\text{R} - \text{MS}_\text{E}}{\text{MS}_\text{R} + (k-1)\text{MS}_\text{E}}$ |
| Two-way consistency average | ICC(3,k) | Fixed | Consistency of average fixed raters | $\dfrac{\text{MS}_\text{R} - \text{MS}_\text{E}}{\text{MS}_\text{R}}$ |
| Pearson correlation | $r$ | N/A | Correlation only (not agreement) | $r = \dfrac{\sum (x_i - \bar{x})(y_i - \bar{y})}{\sqrt{\sum (x_i - \bar{x})^2 \sum (y_i - \bar{y})^2}}$ |

**Notation:**

- $\text{MS}_\text{R}$: Mean square between targets (rows)

- $MS_C$: Mean square between raters (columns)
- $MS_E$: Residual mean square (error)
- $n$: Number of targets
- $k$: Number of raters

**Formulas:**

$$MS_R = \frac{k}{n-1} \sum_{i=1}^{n} (S_i - \overline{X}_{tot})^2$$

$$MS_C = \frac{n}{k-1} \sum_{j=1}^{k} (M_j - \overline{X}_{tot})^2$$

$$MS_E = \frac{\sum_{i=1}^{n} \sum_{j=1}^{k} (x_{ij} - M_j)^2 - k \sum_{i=1}^{n} (S_i - \overline{X}_{tot})^2}{(n-1)(k-1)}$$

$$S_i = \frac{1}{k} \sum_{j=1}^{k} x_{ij}$$

$$M_j = \frac{1}{n} \sum_{i=1}^{n} x_{ij}$$

$$\overline{X}_{tot} = \frac{1}{k \cdot n} \sum_{i=1}^{n} \sum_{j=1}^{k} x_{ij}$$

A.2 CHERNOFF BOUND ON INTRA-CLASS CORRELATION COEFFICIENT

Given the population ICC stated in the previous section, we denote the sample ICC of $n$ samples as $\hat{\rho}_n$ and calculate as the formula listed in our table for $ICC(3,1)$. In Fisher (1925), Fisher demonstrates that with the assumption of sufficiently large number of samples, and given that $\hat{\rho}_n$ is not close to $-1$ nor $1$, the distribution of $\hat{\rho}_n$ on bivariate Gaussian random variables asymptotically approaches Gaussian with parameters $\mathbb{E}[\hat{\rho}_n] = \rho$ and $\text{Var}(\hat{\rho}_n) = \frac{(1-\rho^2)^2}{n-1}$. As earlier work on sample bounds for ICC leverage this approximation and associated assumptions (Zou, 2012), we consider these assumptions and approximations reasonable. As stated in Section 3.2, we assume a bivariate normal distribution for LLM-annotated and human annotated samples.

Thus, we only require a few additional (already extant) propositions to derive an approximate Chernoff bound.

**Proposition 2.** *(Chernoff, 1952) For any random variable $X$, the Chernoff bound dictates that*

$$\Pr(X \geq \varepsilon) \leq \inf_{\lambda \geq 0} \varphi_X(\lambda) e^{-\lambda \varepsilon}$$

*where $\varphi_X(\lambda)$ is the moment generating function for $X$.*

For a Gaussian random variable $X \sim \mathcal{N}(\mu, \sigma^2)$, the moment generating function $\varphi_X(\lambda) = \exp(\mu\lambda + \sigma^2\lambda^2/2)$. Due to linearity of the Gaussian distribution, $X - \mu \sim \mathcal{N}(0, \sigma^2)$, and the moment generating function is $\varphi_X(\lambda) = \exp(\sigma^2\lambda^2/2)$. Thus, the Chernoff bound for a Gaussian random variable is

$$\Pr(X - \mu \geq \varepsilon) \leq \exp\left(-\frac{\varepsilon^2}{2\sigma^2}\right)$$

Analogously,

$$\Pr(\mu - X \geq \varepsilon) \leq \exp\left(-\frac{\varepsilon^2}{2\sigma^2}\right)$$

Therefore, as the events $X - \mu \geq \varepsilon$ and $\mu - X \geq \varepsilon$ are mutually exclusive and constitute the event $|X - \mu|$,

$$\Pr(|X - \mu| \geq \varepsilon) \leq 2\exp\left(-\frac{\varepsilon^2}{2\sigma^2}\right)$$

Combining this with the fact that $\hat{\rho}_n$ approaches Gaussian asymptotically with variance $\frac{(1-\rho^2)^2}{n-1}$, we obtain our desired approximate bound

$$\Pr[|\hat{\rho}_n - \rho| \geq \varepsilon] \lesssim 2\exp\left(-\frac{(n-1)\varepsilon^2}{2(1-\rho^2)^2}\right)$$

As standard with concentration inequalities, we can derive the necessary $n$ such that $|\hat{\rho}_n - \rho| \geq \varepsilon$ with at most probability $\delta$ by setting $\delta$ equal to our bound and solving for $n$.

$$\delta = 2\exp\left(-\frac{(n-1)\varepsilon^2}{2(1-\rho^2)^2}\right)$$

$$\log\left(\frac{2}{\delta}\right) = \frac{(n-1)\varepsilon^2}{2(1-\rho^2)^2}$$

$$\frac{2(1-\rho^2)^2}{\varepsilon^2}\log\left(\frac{2}{\delta}\right) = (n-1)$$

$$1 + \frac{2(1-\rho^2)^2}{\varepsilon^2}\log\left(\frac{2}{\delta}\right) = n$$

### A.3    COMPARISON BETWEEN CONFIDENCE INTERVAL AND CONCENTRATION INEQUALITY BOUNDS

In the context of the bound on a $(1-\alpha)100\%$ confidence interval, as detailed in Section **??**, the derived bound is technically on the confidence interval itself, in the case of (Zou, 2012), a Wald interval, i.e. $z_{\alpha/2}\hat{\sigma}_{\hat{\rho}_n}$. The $(1-\alpha)100\%$ two-sided confidence interval itself is already set such that $\Pr[|\hat{\rho}_n - \rho|] = 1 - \alpha$. Thus, probability of the event $|\hat{\rho}_n - \rho| \geq \varepsilon$ must be decomposed as:

$$\Pr[|\hat{\rho}_n - \rho| \geq \omega] = \Pr[|\hat{\rho}_n - \rho| \geq z_{\alpha/2}\hat{\sigma}_{\hat{\rho}_n} \wedge z_{\alpha/2}\hat{\sigma}_{\hat{\rho}_n} \geq \omega]$$

$$= \Pr[|\hat{\rho}_n - \rho| \geq z_{\alpha/2}\hat{\sigma}_{\hat{\rho}_n}] \cdot \Pr[z_{\alpha/2}\hat{\sigma}_{\hat{\rho}_n} \geq \omega]$$

$$= (1-\alpha) \cdot \Pr[z_{\alpha/2}\hat{\sigma}_{\hat{\rho}_n} \geq \omega]$$

where indeed the latter term is the component bounded in Zou (2012) (and set to $1 - \beta$, with $\beta$ the pre-specified assurance probability). Thus, to translate between the frameworks, $\varepsilon = \omega$ and $\delta = 1 - (1-\alpha) \cdot (1-\beta)$. Note that their analysis involves multiple parameters $(\alpha, \beta)$ for our single $\delta$, and thus is not directly translatable in the opposite direction, as many choices for $\alpha$ and $\beta$ suffice. However, $\alpha = 0.05$ can be assumed to be effectively constant, as this is common practice in applied statistics.

### A.4    SELECTION METHODS

#### A.4.1    RANDOM SELECTION

The baseline random selection strategy serves as our control method:

$$S_{\text{random}} = \text{UniformSample}(\mathcal{N}, k)$$

where items are selected uniformly at random from the full set $\mathcal{N}$ without replacement.

#### A.4.2    STRATIFIED SELECTION

Stratified selection partitions the cheap ratings into $k$ quantile-based strata and selects one representative from each stratum:

$$Q_j = \text{Quantile}(G, \frac{j}{k}) \quad \text{for } j = 0, 1, \ldots, k$$

$$\text{Stratum}_j = \{i \in \mathcal{N} : Q_{j-1} \leq g_i \leq Q_j\}$$

$$S_{\text{stratified}} = \bigcup_{j=1}^{k} \text{UniformSample}(\text{Stratum}_j, 1)$$

### A.4.3   QUERY-BY-COMMITTE (QBC)

This strategy prioritizes items where multiple cheap raters exhibit maximum disagreement, under the hypothesis that such items are most informative:

$$d_i = |g_i^{(1)} - g_i^{(2)}| \quad \text{for } i \in \mathcal{N}$$

$$S_{\text{QBC}} = \arg\max_{|S|=k} \sum_{i \in S} d_i$$

where $g_i^{(1)}$ and $g_i^{(2)}$ represent ratings from two different cheap judges.

### A.4.4   STRATIFIED QBC SELECTION

The hybrid approach combines stratified and disagreement-based selection:

$$S_{\text{sQBC}} = S_{\text{sQBC}}^{(k/2)} \cup S_{\text{QBC}}^{(k/2)}$$

where $S_{\text{strat}}^{(k/2)}$ contains $k/2$ items selected via stratification and $S_{\text{QBC}}^{(k/2)}$ contains the remaining items selected by disagreement, excluding those already chosen.

### A.4.5   CLUSTER-BASED SELECTION

This method applies K-means clustering to identify $k$ clusters in the cheap rating space and selects the item closest to each cluster centroid:

$$\{\mathbf{c}_1, \mathbf{c}_2, \ldots, \mathbf{c}_z\} = \text{KMeans}(G, z)$$

$$S_{\text{cluster}} = \left\{ \arg\min_{i \in C_j} |g_i - c_j| : j = 1, 2, \ldots, k \right\}$$

where $C_j$ represents the set of items assigned to cluster $j$ and $c_j$ is the corresponding cluster center.

### A.4.6   MAXIMIZE-VARIATION SELECTION

Maximize-Variation selection aims to preserve the between-item variance crucial for ICC computation by iteratively selecting items that maximize subset variance:

$$S_0 = \left\{ \arg\min_{i \in \mathcal{N}} \left| g_i - \text{Median}(G) \right| \right\}$$

$$S_{t+1} = S_t \cup \left\{ \arg\max_{i \in \mathcal{N} \setminus S_t} \text{Var}(G_{S_t \cup \{i\}}) \right\}$$

### A.4.7   DENSITY-BASED SELECTION

Density-based selection balances representation between high-density regions (typical cases) and low-density regions (outliers) using kernel density estimation:

$$\rho_i = \text{KDE}(g_i | G) \quad \text{for } i \in \mathcal{N}$$

$$S_{\text{high}} = \text{Sample} \left( \arg\max_{|T|=k} \sum_{i \in T} \rho_i, k/2 \right)$$

$$S_{\text{low}} = \text{Sample} \left( \arg\min_{|T|=k, T \cap S_{\text{high}}=\emptyset} \sum_{i \in T} \rho_i, k/2 \right)$$

$$S_{\text{density}} = S_{\text{high}} \cup S_{\text{low}}$$

where $\text{KDE}(g_i | G)$ represents the kernel density estimate of rating $g_i$ given the distribution of all cheap ratings $G$.

## A.5 Judge Model Information

We use GPT-4o-mini with API version 2023-05-15 as our primary model judge for this setting due to the balance of accuracy and cost. We use a temperature score of 0.2. For QBC based selection methods, we use Claude-3.5-Sonnet as an additional committee member. Future work should involve exploring generalizability of claims across additional judge model architectures and families.

## A.6 Absolute Improvement Between Clustering and Random

We include absolute improvement between clustering selection and random selection to further ground the relative improvement results present above. We see from these results that occasions where random selection outperforms cluster-based selection are outperforming by relatively small margins compared to larger gains when $n_{expensive} \leq 20$.

| $n_{\text{expensive}}$ | HANNA | MedVAL | MSLR | SummEval |
|---|---|---|---|---|
| 10 | 0.073 | 0.026 | 0.014 | 0.051 |
| 20 | 0.031 | 0.028 | 0.010 | 0.010 |
| 30 | 0.013 | 0.021 | 0.014 | 0.002 |
| 40 | 0.011 | 0.016 | 0.011 | 0.000 |
| 50 | 0.010 | 0.006 | 0.005 | 0.000 |
| 60 | 0.000 | 0.004 | 0.001 | -0.004 |
| 70 | 0.002 | 0.005 | 0.002 | -0.007 |
| 80 | 0.002 | -0.003 | 0.000 | -0.003 |
| 90 | 0.004 | 0.002 | 0.003 | 0.000 |

Table 5: Absolute Improvement Results across datasets for varying $n_{\text{expensive}}$.

## A.7 Confidence Interval Visualization

We report the confidence interval width on the returned estimated subset ICC value from cluster-based sampling and random sampling. We use Fisher's Z transformation to derive the 95% confidence interval width, and report the standard deviation of this width across $k = 100$ iterations.

## A.8 Ablation: Different judge model

We flip the main and QBC models to assess the reliance of our results on the specific judge model, such that our main model becomes Claude-3.5-Sonnet and our QBC model becomes GPT-4o-mini. We see that while the exact estimation error varies as a result of the judge, the benefit of cluster-based sampling over random sampling remains consistent.

## A.9 Ablation: Different annotation budget ratio

We see with our full dataset of 300 samples that clustering outperforms random selection uniformly until our data budget is $n_{expensive} = 40$, which is 13% of the dataset. We run an ablation with the full dataset equal to 150 samples such that we can observe whether the trends correspond to the magnitude of the budget or the proportion of the budget in relation to the total dataset size. Similarly, with this subsample, we see that the first instance of random outperforming cluster selection is on the MSLR dataset at N=20, which is 13% of the data. This supports the claim that these methods are best suited for data scarce settings and that data scarcity can be defined in relation to the total data available.

## A.10 Large Language Model Usage

We used large language models during the brainstorming and writing phases of our project. For writing, we used primarily for polishing grammar and improving flow between topics. We additionally used LLMs to recommend citations that may be relevant to our particular work.

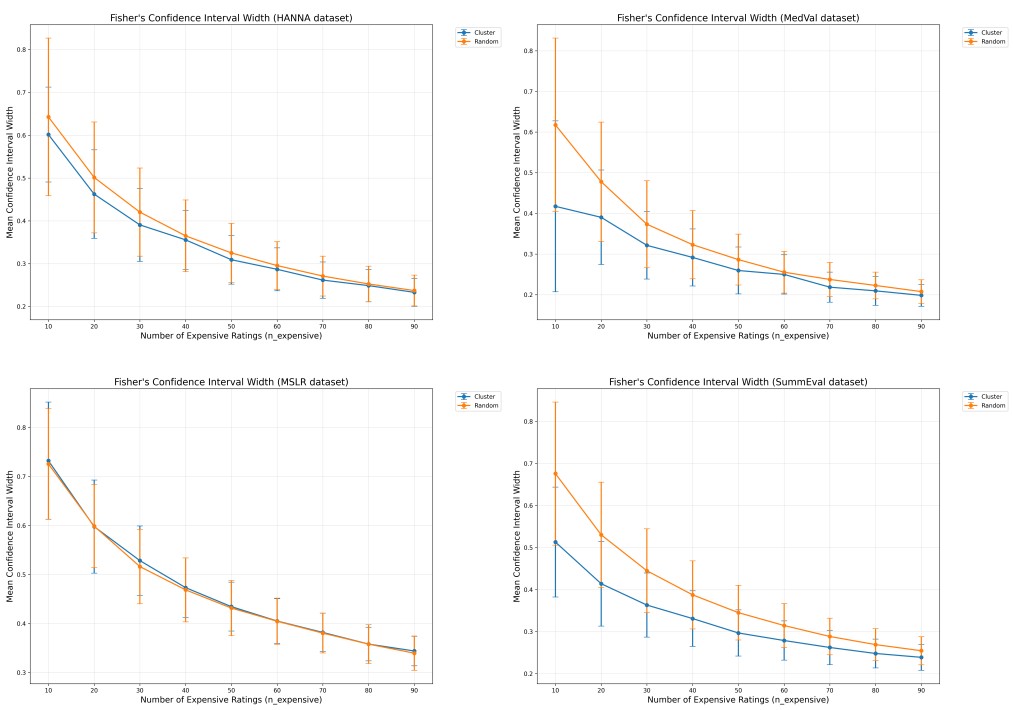

Figure 3: **Cluster selection yields more narrow confidence intervals than random selection.** 95% confidence intervals for intraclass correlation coefficients (ICC) estimated using the Fisher Z transformation.

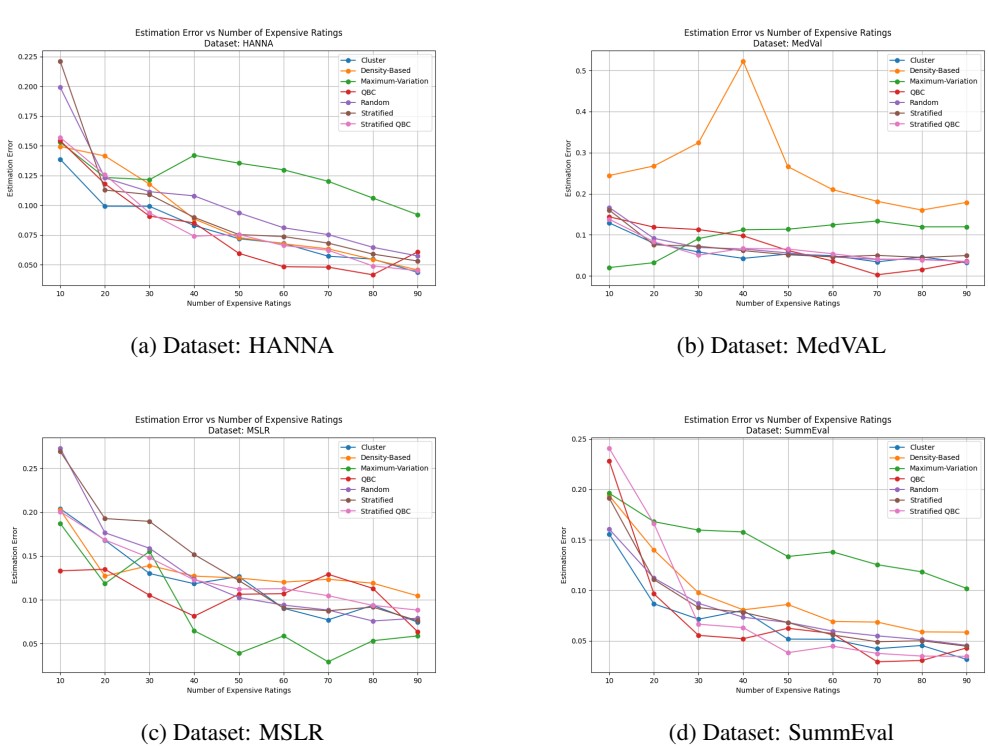

(a) Dataset: HANNA

(b) Dataset: MedVAL

(c) Dataset: MSLR

(d) Dataset: SummEval

Figure 4: **Cluster based sample selection for annotation leads to lower estimation error across models.** We perform a complementary analysis of all selection methods with Claude-3.5-Sonnet as the judge model and observe consistency in trends, with cluster based selection still being a consistent method for improving performance over random.

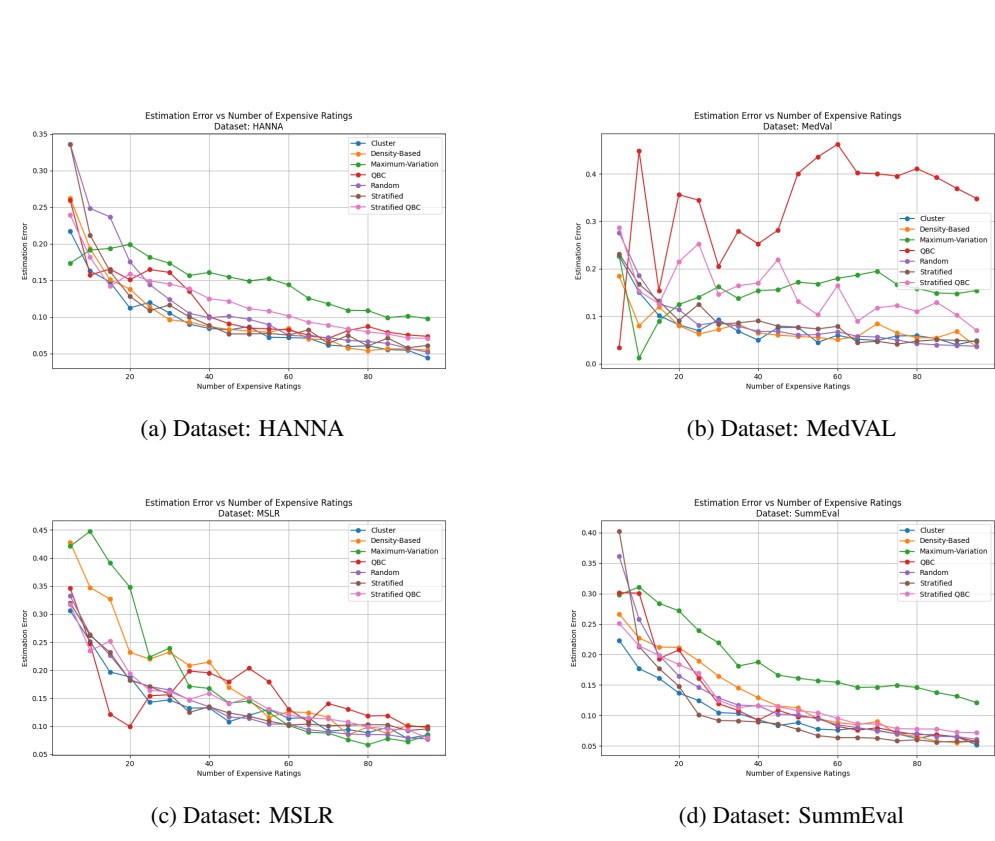

(a) Dataset: HANNA

(b) Dataset: MedVAL

(c) Dataset: MSLR

(d) Dataset: SummEval

Figure 5: **Cluster based selection advantages scale with dataset size.** We evaluate our selection strategies with N=150 such that all $n_{expensive}$ represent a larger fraction of the sample population and observe that general trends hold as with N=300, specifically that cluster selection leads to monotonic improvement until 13% of data is sampled and then improvement starts to converge with random performance.

