# OpenReview forum: "SMARTER SAMPLING FOR LLM JUDGES: RELIABLE EVALUATION ON A BUDGET"
_ICLR.cc/2026/Conference — ICLR 2026 Conference Withdrawn Submission_

### Official Review · Reviewer_R9Sw · 2025-10-18

**Soundness:** 3
**Presentation:** 3
**Contribution:** 3
**Rating:** 6
**Confidence:** 3

**Summary:**

This paper studies how to evaluate the reliability of LLM judges with minimal human annotation, a task that has broad applications when human labels are costly to obtain. The authors derive a Chernoff-style bound on the estimation error of the intra-class correlation (ICC), which provides practical guidance on the required number of human annotations. The proposed sampling strategies are demonstrated on several real-world datasets.

**Strengths:**

1. This paper, with a clear motivation, is well written and well organized.
2. The problem studied in this paper is of importance across fields. This paper balance the theoretical guarantee with practical considerations.
3. Extensive empirical results are presented to demonstrate the effectiveness of proposed approach.

**Weaknesses:**

1. Notations in Section 3.1 are not fully self-contained. For example, the variance components are not explicitly defined in this section. It would be better to add a subsection to introduce necessary notations.

2. To understand the intuition behind the definition of \(S*\), I was wondering if $\hat \rho$ satisfies the following property: for any $S\_1 \subseteq S\_2$, it holds that $\hat \rho(H_{S\_1},G_{S\_1}) \leq \hat \rho(H_{S\_2},G_{S\_2})$. If this property is not true, it is unclear why \(S*\) defined in this way would be the correct target.

3. With sampling strategies in Table 3, the effective samples are actually obtained by searching over the entire sample space. I wonder if there are active leaning based sampling strategies that will not look at the entire sample space. It would be super helpful if authors can elaborate more on this part.

I am happy to increase my score if my questions can be addressed.

**Questions:**

Please the section of weakness.

---

> ### Author Response · Authors · 2025-12-01
>
> We thank the reviewer for the constructive comments on our paper. We respond to the listed weaknesses as follows:\
> \
> **W1:** Thank you for the note on Section 3.1 notation. The variance components parameterize the distributions of the respective additive components of the random effects model and are assumed to be unknown. Nonetheless, we appreciate that the random effects model and its connection to ICC could be given a more thorough background in the main paper, and will do so in further writing.\
> \
> **W2:** In terms of the intuition behind S*, we want a set of size b < n, where n is the size of the full dataset, for which the empirical ICC on S* most closely approximates the population ICC. We note in footnote 2 that we consider phat_n(H, G) to roughly be the population ICC. Thus, we aim to minimize the difference between phat_b(H_S, G_S) and phat_n(H, G). We are unsure why exactly that would connect to the reviewer’s comment, but perhaps the confusion is over what is being minimized – in this case, it is not the size of the set. Alternatively, perhaps the reviewer means that given any S1 in S2, it holds that |phat(Hs1, Gs1) - p| >= |phat(Hs2, Gs2) - p|. This could perhaps be said to be true in expectation over random draws of S2 and S1, due to the law of large numbers, however we are trying to choose S* in a principled manner, not over random draws, and the size of S* is fixed.\
> \
> **W3:** Thank you for this interesting point regarding active sampling strategies that don’t search over the entire sample space. While this raises interesting directions regarding future methodologies that are increasingly efficient, we respectfully propose that the usage of the full sample space is a strength of the sampling strategies in Table 3. Using these different methods, we are able to make more informed decisions regarding the marginal distribution of the judge labels and how these may relate to the joint distribution between the judge and the human evaluators. Further, in the specific setting that we explore, we do not consider active learning approaches as many annotation efforts by true practitioners require single elicitation structure from humans and as such, we wish to capture the true alignment between humans and automated judges in such a way as if there were more human annotations.

---

### Official Review · Reviewer_XB84 · 2025-10-30

**Soundness:** 2
**Presentation:** 3
**Contribution:** 2
**Rating:** 4
**Confidence:** 2

**Summary:**

This paper focuses on estimating the Intraclass Correlation Coefficient (ICC) between LLM judges and humans, aiming to answer two core questions: how many human annotations are needed for accurate ICC estimation, and how to select informative samples to minimize annotation costs. Centered on the estimation of the Intraclass Correlation Coefficient (ICC): Theoretically, a concentration inequality based on the Chernoff bound is derived, reducing the required sample size by an average of 18% compared to the baseline method（2012）.

**Strengths:**

* The study brings a fresh angle by treating the selection of annotation samples for LLM judges as a core-set selection task. It also cleverly mixes classic statistical methods (Chernoff bounds) with clustering and active learning concepts to make ICC estimation more efficient with limited resources. This combination hasn't really been tried before for evaluating LLM judge reliability on a tight budget.
* The empirical design is relatively rigorous, with evaluations across 4 diverse real-world datasets (covering 15 assessment axes) and 100 rollouts to reduce random variance.

* Overall Well-Written: The paper is generally well-written.

**Weaknesses:**

* Unverified bivariate normality assumption:​​ The derivation requires LLM/human scores to follow a bivariate normal distribution, but this assumption remains untested across the four datasets. The study fails to address how skewed real-world ratings or alternative distributions (e.g., Poisson/uniform) affect the Chernoff-bound error probability, or whether more robust assumptions (sub-Gaussian) are needed.


* Unclear large-sample threshold：The derivation relies on "sufficiently large n for CLT" but only vaguely mentions n<30 violating assumptions, whether datasets with different ICC values (MSLR's 0.346 vs. MedVAL's 0.716) require varying sample sizes.


* Unresolved prior ICC (ρ) requirement:​​ The sample size formula requires knowing the population ICC (ρ), but ρ is the unknown target to be estimated via human annotation. Although the paper later uses LLM scores as a proxy to estimate ρ, it fails to address how the estimation error in this proxy ρ affects the sample size calculation.

**Questions:**

Please see the weaknesses.

---

> ### Author Response · Authors · 2025-12-01
>
> Thank you to the reviewer for their suggestions and constructive concerns about the paper! We discuss the weaknesses in the following points:\
> \
> **W1:** We agree with the reviewer that the theoretical assumptions are limiting in practice, but refer to previous work that relies on identical assumptions, such as Bonett et al. 2002, Zou et al. 2012, (cited in the paper), and believe these assumptions to largely be standard. We thank the reviewer for the suggestion of investigating the effects when these assumptions are violated in practice and will consider it for further experimentation. Some of the suggested distributions, such as Poisson, would not apply in our case due to the time component of Poisson distributions. We note however, that the bivariate Gaussian assumption on the LLM/human scores is more an assumption required in order to approach a nicely parameterized normal distribution of ICC values, with known variance and mean (Fisher 1925); in other cases, the distribution has not been described and like Pearson’s correlation coefficient, likely has no closed-form expression. \
> \
> **W2:** As in the above paragraph, the sufficiently large n for CLT is solely needed such that the distribution of ICC values roughly approaches a normal distribution (as in Zou 2012). Given that, the central limit theorem is generally considered to require roughly n >= 30, but the population ICC is irrelevant to this consideration.\
> \
> **W3:** We acknowledge the existence of this estimation error in the paper. We thank the reviewer for the suggestion and will perform more sensitive analysis in estimation error as suggested. Nonetheless, we note that previous work in practical statistical applications of ICC bounds and distribution parameterization rely on the same estimation and thus investigation in similar work already exists (Fisher 1925, Zou 2012).

---

### Official Review · Reviewer_UiqQ · 2025-10-31

**Soundness:** 3
**Presentation:** 2
**Contribution:** 2
**Rating:** 4
**Confidence:** 4

**Summary:**

This work focuses on how to estimate the alignment of the LLM judgement with human annotations under limited budgets. To archive this goal, the authors first develop a theoretical framework based on Chernoff bound to derive a lower bound on the human annotation sample size which can be used to estimate the ICC (which measures the human-LLM judges alignment) reliably. Then the derived bound is used to empirically select most effective sampling method for ICC estimation. The experimental results shows that the cluster-based sampling consistently yields lowest estimation error.

**Strengths:**

1 A theoretical framework with thorough analysis for the human-LLM judges alignment problem is provided in the paper, and serves as a theoretical foundation for the proposed core-set sampling problem in the experiment.
2 The formulations and derivations are logically sound and clearly expressed.
3 The paper is well-organized and the texts are generally easy to understand.

**Weaknesses:**

1 The title of the paper is a bit misleading. "Smarter Sampling" seems like a novel sampling method. Instead, what the paper actually proposed is a way to select one candidate method among **existing** sampling methods.
2 The aim of this paper is to improve the reliability of LLM evaluation systems. But little detail is given on which specific LLM is used as judge model in the main paper. The appendix A.5 mentions GPT-4o-mini and Claude-3.5-Sonnet, but the judgement capabilities of different models is also an important factor in LLM evaluation systems. Maybe the paper should estimate the ICCs of more judge models with varying sizes and capabilities.
3 I notice a gap between the theoretical sections and the experiment sections. The lower bound can be estimated and approximately calculated from the inequality at line 248 and the results are given in Table 1. But in the experiment sections, the actual sampling sizes are still empirically selected from a range of candidates.
4 Minor: The paper lacks equations reference number, making it difficult to navigate.

**Questions:**

please refer to weakness section.

---

> ### Author Response · Authors · 2025-12-01
>
> We thank reviewer UiqQ for their helpful comments engaging with our work! We engage with your feedback directly below:\
> **W1: Limited judge models in empirical evaluations**\
> We thank you for this important point and agree that additional judge models would help to show generalizability. We show that across 2 different frontier LLM judges (both of which can reasonably be assumed to be used in deployment settings) and across 4 datasets with 15 independent sets of annotations, cluster-based sampling outperforms random selection at low sample budgets. We will include additional evaluation across judge model sizes and capabilities in future iterations of this work.
>
> **W2: Gap between theory and empirical results**\
> We appreciate this feedback and highlight that both the theory and the empirical results aim to address the same core motivation: how do we increase trust in our LLM-Judge system? The theoretical bounds highlight our capacity to calculate ICC at lower sample allocations than previously recorded, while the empirical approaches highlight that further gains can be achieved (albeit with more limited theoretical guarantees) when using the LLM judge answer distribution to select the next data points for annotation. In future work, we will incorporate additional experiments and discussion to bridging this gap and making the connection between our empirical results and theoretical guarantees more clear.

---

### Official Review · Reviewer_b9C4 · 2025-11-03

**Soundness:** 3
**Presentation:** 2
**Contribution:** 2
**Rating:** 2
**Confidence:** 3

**Summary:**

This paper tackles the problem of estimating how many human annotations are needed to estimate agreement between human raters and an LLM “judge”, and which examples should be selected for annotation under a limited budget. The authors focus on the intra-class correlation coefficient (ICC) as the reliability metric and study both (i) sample size calculation and (ii) sampling strategies. The overall idea of automating annotation planning, both estimating the required sample size and selecting which items to annotate, is well-motivated and practically useful, particularly given the growing reliance on LLM judges and the high cost of expert labels.

**Strengths:**

- The study drives a Chernoff bound for ICC based on Fisher’s asymptotic normal approximation, which yields a lower bound on the number of human annotations required to guarantee that the empirical ICC is within ε of the population ICC with high probability.

- They compare six sampling strategies for selecting instances to annotate (including clustering-based, stratified, and random) across four datasets and multiple evaluation axes, and report that clustering-based selection can improve ICC estimation precision.

**Weaknesses:**

- My main concern is that the reliability and robustness of these sample-size estimates and sampling strategies are likely to be highly task- and model-dependent. The bound is based on assumptions such as approximate normality, sufficiently large n, and ICC values not too close to the boundaries, and it is not entirely clear how sensitive the resulting recommendations are when these assumptions are violated in practice (e.g., skewed rating distributions, heavy tails, or low human–LLM alignment).

- The empirical study is conducted on a limited set of datasets and judge models, which all have reasonably high ICC in several cases;

**Questions:**

Can you provide results in settings where the LLM judge is poorly aligned with humans? The cheap LLM scores may be a weak proxy for the human variance structure, and the proposed clustering-based selection may behave much closer to random.

---

> ### Author Response · Authors · 2025-12-01
>
> We thank reviewer b9C4 for their helpful comments. We aim to clarify the following points below:
>
> **W1: Limiting theoretical assumptions** \
>  We agree with the reviewer that the theoretical assumptions are limiting in practice, but refer to previous work that relies on identical assumptions, such as Bonett et al. 2002, Zou et al. 2012, (cited in the paper), and believe these assumptions to largely be standard. We thank the reviewer for the suggestion of investigating the effects when these assumptions are violated in practice and will consider it for further experimentation. However, we do note that low human-LLM alignment does in fact satisfy our assumptions – specifically, low human-LLM alignment would correspond to an ICC value near 0, which is not near the boundaries of the domain ($\rho \in [-1, 1]$).
>
> **W2: Empirical study is limited on datasets and judge models which have high ICC**\
> We thank the reviewer for this point regarding the limitations of the empirical experiments. We strive to thoroughly evaluate across different datasets with different ICC scores. The ICC of humans and model judges for MSLR datasets are 0.346, while the ICC of HANNA, MedVAL, and SummEval, respectively, are 0.655, 0.716, 0.627.  We show with our MSLR experiments that although improvements over random with cluster selection are more pronounced in higher ICC settings, results are still better than random even in low ICC settings.

---

### Note · Authors · 2025-12-01

**Comment:**

We sincerely thank all reviewers for their thoughtful engagement. Reviewers appreciated our novel theoretical framework deriving Chernoff bounds for ICC estimation, noting that our approach of treating annotation sample selection as a core-set selection task represents a fresh angle that "hasn't really been tried before for evaluating LLM judge reliability." They recognized the clear motivation and importance of the problem, the rigor of our empirical design across 4 diverse datasets with 100 rollouts, and found our formulations logically sound and clearly presented.

However, after careful consideration, **we have decided to withdraw this submission to strengthen our work in three key areas**: (1) expanding empirical evaluation across more judge models and diverse ICC settings, (2) clarifying the connection between our theoretical guarantees and empirical sampling strategies, and (3) investigating robustness when theoretical assumptions are violated in practice. We believe these revisions will address reviewer concerns while enhancing the core strengths they identified, resulting in a significantly stronger contribution. We look forward to resubmitting a more comprehensive version in the future.

**Withdrawal Confirmation:**

I have read and agree with the venue's withdrawal policy on behalf of myself and my co-authors.